# Epidemiology of Cerebral Palsy among Children and Adolescents in Arabic-Speaking Countries: A Systematic Review and Meta-Analysis

**DOI:** 10.3390/brainsci12070859

**Published:** 2022-06-29

**Authors:** Sami Mukhdari Mushta, Catherine King, Shona Goldsmith, Hayley Smithers-Sheedy, Al-Mamoon Badahdah, Harunor Rashid, Nadia Badawi, Gulam Khandaker, Sarah McIntyre

**Affiliations:** 1The Children’s Hospital at Westmead Clinical School, Faculty of Medicine and Health, The University of Sydney, Westmead, NSW 2145, Australia; smus9514@uni.sydney.edu.au (S.M.M.); harunor.rashid@health.nsw.gov.au (H.R.); 2Public Health Authority, Riyadh 13354, Saudi Arabia; 3Sydney Institute for Infectious Diseases, Faculty of Medicine and Health, The University of Sydney, Westmead, NSW 2145, Australia; catherine.king@sydney.edu.au; 4Cerebral Palsy Alliance Research Institute, Specialty of Child & Adolescent Health, Sydney Medical School, Faculty of Medicine & Health, The University of Sydney, Camperdown, NSW 2050, Australia; sgoldsmith@cerebralpalsy.org.au (S.G.); hsmitherssheedy@cerebralpalsy.org.au (H.S.-S.); nadia.badawi@health.nsw.gov.au (N.B.); smcintyre@cerebralpalsy.org.au (S.M.); 5Department of Family and Community Medicine, Faculty of Medicine in Rabigh, King Abdulaziz University, Jeddah 22252, Saudi Arabia; ambadahdah@kau.edu.sa; 6Grace Centre for Newborn Intensive Care, Sydney Children’s Hospital Network (SCHN), Westmead, NSW 2145, Australia; 7School of Health, Medical and Applied Sciences, Central Queensland University, Rockhampton, QLD 4701, Australia; 8Central Queensland Public Health Unit, Central Queensland Hospital and Health Service, Rockhampton, QLD 4700, Australia

**Keywords:** Arab, Arabic-speaking countries, children, adolescents, cerebral palsy, epidemiology

## Abstract

Background: Studies on cerebral palsy among children and adolescents in Arabic-speaking countries are scarce. In this systematic review, we aimed to describe the epidemiology of cerebral palsy among children and adolescents in Arabic-speaking countries in terms of prevalence, risk factors, motor types, and rehabilitation. Methods: Six key bibliographic databases were searched for relevant literature published to 17 July 2021. Titles and abstracts were screened for potential inclusion and two independent reviewers screened the full texts of potential articles following pre-defined inclusion/exclusion criteria. The included studies were evaluated independently by three reviewers. The risk of bias was assessed, and data were extracted and analysed. Results: A total of 32 studies from 7 countries met our inclusion criteria. The prevalence of cerebral palsy in Arabic-speaking countries was 1.8/1000 live births (95% CI: 1.2–2.5). Spastic cerebral palsy was the most common motor type, representing 59.8% (95% CI: 46.2–72.7) of pooled estimates. This included children with spastic quadriplegia, diplegia, and hemiplegia; 25.1% (95% CI: 18.2–32.8), 16.2% (95% CI: 11.4–23.3), and 10.4% (95% CI: 7.3–13.8), respectively. Consanguinity was high and represented 37.7% (95% CI: 29.3–46.6). Only one included study reported the types of rehabilitation received (e.g., physiotherapy and assistance devices). Conclusions: This paper provides a summary of the epidemiology of cerebral palsy in Arabic-speaking countries and highlights areas for future research. There is still a substantial knowledge gap on the epidemiology of cerebral palsy in these regions. Countries in the Arab region should follow examples of countries that have successfully established cerebral palsy registries to generate evidence on epidemiology of cerebral palsy and opportunities for prevention.

## 1. Introduction

Cerebral palsy (CP) is a term used to describe a group of lifelong disorders of mobility and posture that limit activities and are caused by nonprogressive disturbances in the developing foetus or infant brain. The motor disorders of CP are frequently accompanied by sensory, perception, cognitive, and communication disturbances, as well as epilepsy and subsequent musculoskeletal problems [1]. The overall prevalence of CP has remained stable at around 2.1 per 1000 live births [2]; however, it is lower in high-income countries (HICs) where the prevalence is declining with ranges from 1.4 to 1.7 per 1000 live births [3,4,5,6].

CP is commonly classified based on motor type (spasticity, dyskinesia, and ataxia) and by spastic topography (e.g., quadriplegia, diplegia, and hemiplegia) [7]. Spastic CP is the most common type that affects 70–80% of individuals [8]. There are also other classification methods to describe the functional motor severity of CP. Examples are the Gross Motor Function Classification System (GMFCS), the Manual Ability Classification System (MACS), and the Communication Function Classification System (CFCS) [9,10,11,12].

The causal pathways to CP are complex and not completely understood. The timing of brain injury or maldevelopment is generally classified as pre/perinatal, which accounts for 85–90% of CP, or post-neonatally acquired (i.e., >28 days and prior to 2 years of age [13,14]. Pre/perinatal risk factors for CP include a very-low-birthweight, congenital infections such as TORCH (*Toxoplasma gondii*, other agents, rubella, cytomegalovirus, and herpes simplex), cerebral thrombosis, placental abnormalities, smoking, trauma, prematurity, multiple gestation, and genetic risk factors. Post-natal risk factors include, but are not limited to postnatal infections, accidental, and non-accidental injuries [13,14,15].

The Arabic-speaking countries (ASCs) are countries in the Middle East and North Africa, consisting of 22 countries, of which Arabic is an official language. Approximately 392.4 million people live in this region in an area of over 13 million km^2^ [16,17]. These countries share many similarities, for instance, endogamy is a part of culture in these countries, this is important to know as consanguinity is a potential risk factor for CP [18,19]. However, they differ in many respects including their political and cultural landscapes, and their economic classifications, which range from high-income (HIC), upper-middle income (UMIC), lower-middle income (LMIC), and low-income countries (LIC) [19,20]. Table 1 provides a brief overview.

There is a significant lack of information about the epidemiology of CP in the ASCs in terms of prevalence rates, types, causes, and/or risk factors. This information is needed to inform the development of prevention strategies and to support the planning of services. We aimed to collate the scattered available information on CP epidemiology in the ASCs and highlight any knowledge gaps to pave the way for researchers to bridge these existing gaps.

## 2. Materials and Methods

We conducted this review according to the Preferred Reporting Items for Systematic Reviews and Meta-Analysis (PRISMA) guidelines on conducting systematic reviews, including the 27-item checklist [23,24]. The review protocol was registered in the National Institute for Health Research (NIHR) and the International Prospective Register of Systematic Reviews (PROSPERO) (CRD42020172159) [25].

### 2.1. Data Sources and Search Strategy

An experienced information specialist (C.K.) performed bibliographic database searches to locate publications on Arab children and adolescents with CP. The following databases were searched: OVID Medline All, including Epub Ahead of Print, In-Process and Other Non-Indexed Citations, Daily and Versions (1946–9 July 2021), OVID Embase Classic and OVID Embase (1947–15 July 2021), CINAHL via EBSCO (1982–July 2021), Cochrane Library databases, including the Database of Systematic Reviews (Issue 7 of 12 July 2021) and the Central Register of Controlled Trials (Issue 7 of 12 July 2021), and SCOPUS (1788–17 July 2021) to find publications on CP in Arabic-speaking countries. The final search was completed on 17 July 2021.

The search used database thesaurus terms including ‘Cerebral Palsy’, ‘Epidemiology’, ‘Hospitalization’, ‘Morbidity’, ‘Mortality’, ‘Death’, ‘Incidence’, ‘Prevalence’, ‘Disease Notification’, ‘Diagnosis’, ‘Prognosis’, ‘Infant’, ‘Child’, and ‘Adolescent’. Individual country terms were derived from those listed in Table 1. Relevant text word terms were also used throughout the search strategy. There were no restrictions on publication date or language. Appendix A contains the complete Medline search strategy.

### 2.2. Study Selection

Using the EndNoteX9 citation manager, deduplication was performed and duplicate abstracts from the literature searches removed. A manual re-check was completed to verify deduplication. One author (S.M.M.) screened abstracts and found potential papers that met the inclusion criteria. Two reviewers (S.M.M. and S.M.) independently reviewed full-text versions of these papers using predefined inclusion and exclusion criteria. Any differences were addressed by consensus and consultation with a third reviewer (G.K.).

The inclusion criteria were as follows: (1) original studies, epidemiological studies, observational studies, case-series studies, or chart reviews; (2) the study population of children, adolescents, or both, no older than 18 years of age, with CP; (3) papers that reported CP prevalence, risk factors, and/or rehabilitation services; and (4) papers with full text available.

We excluded papers that: (1) were single case reports, literature reviews, and systematic reviews; (2) included adults, children who did not have CP; (3) were not in ASCs, and (4) were only abstracts such as conference papers/posters. Full-text with no relevant data, insufficient information about CP, high risk of bias, and papers with the same study population. See Figure 1 and Appendix A.

Some papers that discussed CP along with other conditions were included, provided that there were sufficient CP data, and we were able to extract CP specific results to address our research question(s).

### 2.3. Quality Assessment (Risk of Bias Assessment)

Each included article was independently assessed by two reviewers (S.M.M. and one of S.M., S.G., or H.S-S) for risk of bias (ROB) using the Newcastle–Ottawa Quality Assessment Scale (NOS) for case-control, cross-sectional, and cohort studies [26], and a unique quality assessment tool for case-series studies [27]. We used a modified Excel worksheet designed by one reviewer (S.M.) for this purpose. Any discrepancies between the reviewers’ assessments were reviewed and resolved unanimously. Appendix A summarises the assessment results. Publications with an unacceptably high ROB (i.e., overlap between participants in the study, data does not seem to be reported specifically for people with CP, lack of clear definitions, or displayed some discrepancies in numbers) were excluded. A total score of 3 stars or more were accepted as low ROB. See Appendix A for more details.

### 2.4. Data Extraction

Data were extracted from all included studies using a Microsoft Excel template, which was developed by two reviewers (S.M.M. and G.K). One reviewer (S.M.M.) extracted data from all included studies, then the abstracted data were reviewed independently by another reviewer (G.K.).

### 2.5. Data Analysis

A descriptive analysis was completed using Microsoft Excel 365. Tables were created to display descriptive data (e.g., study characteristics and outcome measures) and results. Percentages were rounded to the nearest decimal. In addition, forest plots and funnel plots were created to represent the meta-analysable data with 95% confidence intervals (CI) using MedCalc^®^ Statistical Software version 20.110 [28]. A random effects model was utilised in the analysis to deal with the universe of population included in this paper. Heterogeneity was considered mild if *I*^2^ < 30%, moderate if *I*^2^ = 30–50%, and notable if *I*^2^ > 50%. Moreover, we calculated non-reported percentages and numbers either directly or inversely.

For prevalence, we screened studies that reported prevalence and examined them separately for possible pooling in a meta-analysis. To overcome heterogeneity, we used a random effect model in a meta-analysis. The pooled prevalence was calculated per 1000 live births.

Classification of CP motor types was adapted to match available data. Spastic included children with spastic CP, including all spastic topographies: quadriplegic, diplegic, and hemiplegic. Other motor types reported were dyskinetic/athetoid, hypotonic/atonic, ataxic motor types, and ‘mixed motor types’. We used ‘unclassified’ where motor type was not reported. We were also able to analyse the types of CP based on motor severity using GMFCS; with levels collapsed into two groups: I–III and IV–V [29].

For risk factor analysis, we extracted all risk factors reported in studies. Counts and percentages were undertaken of all risk factors. This list included the following: preterm, perinatal injuries/trauma, jaundice, birth asphyxia, congenital infections TORCH, acquired infections, multiple birth, birth weight less than 2500 gm (including low-birth weight (LBW), very-low-birth weight (VLBW), and extremely-low-birth weight (ELBW)), medical condition of mother, family history of CP, and admission to neonatal intensive care unit (NICU admission). We included the term ‘other’ as many included studies used this term to accommodate other causes that eventually led to children with CP. We analysed consanguinity separately and ran a random-effects model in meta-analysis to assess and overcome inconsistency between the studies.

## 3. Results

### 3.1. Study Selection and Eligibility

A total of 702 records were identified through database searching. After deduplication and a manual re-check, 353 primary titles were screened, of which 279 were excluded. Out of the 74 articles that were included for full-text review, 42 articles were excluded and 32 included for qualitative synthesis. Details are shown in Figure 1.

### 3.2. Study Characteristics of Included Publications

The 32 included studies were published between 1984 and 2020. Most of the studies were from Saudi Arabia (*n* = 10) [29,30,31,32,33,34,35,36,37,38], Jordan (*n* = 8) [39,40,41,42,43,44,45,46], Egypt (*n* = 6) [47,48,49,50,51,52], and Iraq (*n* = 4) [53,54,55,56]. Other studies were from Sudan (*n* = 2) [57,58], Libya (*n* = 1) [59], and Palestine (*n* = 1) [60]. There were a variety of study designs. Most were of case-series (*n* = 13) [32,33,34,36,38,40,41,42,43,44,53,55,56] and cross-sectional design (*n* = 11) [29,31,39,45,46,47,48,49,50,51,52]. Other study designs included: case-control (*n* = 5) [35,54,58,59,60], retrospective review (*n* = 2) [37,57], and prospective/retrospective cohort (*n* = 1) [30]. Seven studies [31,46,47,48,49,50,52] reported comparable prevalence; however, only one study reported birth years [52].

There was diversity among the studies in terms of the setting (e.g., hospitals, rehabilitation centres, and community) and the duration of the study. The pooled total of 3943 children with CP included in the studies ranged from 8 to 475 participants with ages ranging from newborn to 18 years old. Numbers/percentages on sex were available for all studies, which ranged from 35.0% to 70.8% for male, and from 29.2% to 65.0% for female. The detailed characteristics of the included publications are summarised in Table 2.

### 3.3. Prevalence of CP in ASCs

The seven studies that were eligible for analysis were from Egypt, Saudi Arabia, and Jordan, which are classified as LMIC, HIC, and UMIC, respectively. Four studies were community-based and the remaining were hospital-based (i.e., rehabilitation clinics). We acknowledge that care must be taken when interpreting our pool results, as we included data from both hospital and community-based studies, due to the availability of data reported about prevalence. However, this was important as to elaborate the possible estimation from the available data. Out of a total *n* = 3943 children with CP in the included studies in this review, *n* = 725 were eligible for inclusion in the estimation of the pooled prevalence. The prevalence rate of CP was reported in *n* = 7 studies [31,46,47,48,49,50,52]. The hospital-based estimates ranged from 0.6 to 1.4 per 1000 (pooled estimates 0.98 per 1000; 95% CI: 0.60–1.47) and the community-based estimates ranged from 2.04 to 3.6 per 1000 (pooled estimates 2.63/1000; 95% CI: 2.0–3.35). The pooled estimate of all CP birth prevalences was 1.8 per 1000 live births (95% CI: 1.1–2.5). Figure 2 shows the weight of each individual study included in the meta-analysis.

### 3.4. Motor Types and Severity of CP in ASCs

Of the *n* = 32 included papers, *n* = 28 articles reported the CP motor type. Although the classification of motor type differed between articles, we were able to harmonise these classifications. Only four included articles did not report CP types [27,30,31,60]. From the pooled estimates of 3632 children with CP, the most common motor type was spastic CP (59.8%, 95% CI: 46.2–72.7, *p* < 0.0001) and the most common spastic topographies were spastic quadriplegia (25.1%, 95% CI: 18.2–32.8, *p* < 0.0001) and spastic diplegia (16.9%, 95% CI: 11.4–23.3, *p* < 0.0001). See Table 3.

Table 4 reports findings from papers that included data pertaining to motor severity using the GMFCS classification. Meta-analysis of pooled 1107 CP cases suggested that 50.6% of children were described as GMFCS I-III (95% CI: 42.8–58.3, *p* < 0.0001) while 48.5% were described as GMFCS IV-V (95% CI: 41.0–56.0, *p* < 0.0001). Figure 3 and Figure 4 show the results of meta-analyses of the severity of CP among children and adolescents in ASCs based on GMFCS levels. Forest plots show the pooled proportions of severity levels while funnel plots suggested that there was no significant publication bias in the meta-analyses for the proportions of GMFCS (*p* = 0.0833 and *p* = 0.0478 for GMFCS I-III and I-V, respectively).

### 3.5. Risk Factors of CP in ASCs

A total of *n* = 26 studies were eligible for the meta-analysis of risk factors of CP. There were *n* = 3440 (30.7%) cases without a reported risk factor (95% CI: 20.3–42.2, *p* < 0.0001). The majority of children had one or more identified risk factor including birth complications such as birth asphyxia (16.0%, 95% CI: 8.8–24.9, *p* < 0.0001), prematurity/preterm (12.2%, 95% CI: 6.9–18.7, *p* < 0.0001), and a low birth weight (9.7%, 95% CI: 4.4–16.8, *p* < 0.0001), Table 5.

### 3.6. Consanguinity as a Risk Factor for CP in ASCs

Twelve articles were eligible for the meta-analysis of consanguinity. The pooled estimates of 1665 children with CP showed that 37.7% of children (95% CI: 29.3–46.6, *p* < 0.0001) were born from consanguineous marriages. Details are shown in Figure 5.

### 3.7. Rehabilitation Status

Only one study [56], which described 100 children with CP in Iraq, reported rehabilitation received. Physiotherapy was received by 34% of children. While an additional eight studies (*n* = 1084 children) reported that rehabilitation centers were used for recruitment, no details about the types of rehabilitation were provided.

## 4. Discussion

This systematic review is the first to collate and report on the epidemiology of CP among children and adolescents in ASCs. It provides an important overview of CP in Arab societies. CP is a life-long condition for which there are opportunities for prevention in the pre, peri, and postnatal periods [61,62,63].

Globally, CP prevalence ranges from 1 to nearly 4 per 1000 live births [64,65]. The prevalence in our review falls within this range (1.8 per 1000 live births, 95% CI: 1.2–2.5), however, we cannot generalise this prevalence to all ASCs because of the small number of studies included in the meta-analysis, and as most of them were from one country (i.e., *n* = 5 from Egypt, *n* = 1 from Saudi Arabia, and *n* = 1 from Jordan). There is variation in the methodology (i.e., community-based versus hospital-based) and the economic profile of these countries (five from LMIC, one from HIC, and one from UMIC). This is reflected in the range of prevalence reported (0.62 to 3.60 per 1000 LB). This gives us an indication of the range of CP birth prevalence across the region, but more research is required.

Our review suggests that the most common type of CP in ASCs is spastic CP. This is consistent with the CP population globally, however, the proportion of spastic CP differs (59.8% of all spastic CP in our review versus the known percentage of spastic CP that is 85–90%) [8]. Differences in the proportions of CP motor types and spastic sub-types likely reflect the study methodologies and the individual meta-analyses conducted. While the proportions of children described as GMFCS IV-V were high in this review compared to HIC, they are consistent with studies describing CP in other LMICs and UMICs [66,67,68].

In our review, we found that 30.7% children had no identified/reported risk factors (i.e., unknown risk factors) for CP. However, this is expected as the causal pathway to CP often remains unknown when genetic abnormalities, congenital brain deformities, maternal illnesses or fevers, or foetal damage are all possible causes [69]. Birth asphyxia represents 16.0% of the causes of CP in Arab children. This percentage, however, is close to percentages reported in two studies that have been conducted in two western countries [70,71]. NICU admissions in our review accounted for only 2.5% of total cases, which is an extremely low rate, likely reflecting the limited availability or capacity of NICU facilities in many regions [72]. Similarly in this study, preterm birth as a risk factor was low compared to data from other HICs, but was in line with data from Bangladesh, an LMIC [66]. This may be accounted for by a survival bias, with vulnerable preterm babies not surviving to be diagnosed with CP.

We found that 37.7% of children included in articles were born from consanguineous marriages. This is not unexpected as the Arabic culture plays a significant role in endogamy. In most, if not all, Arab societies, consanguineous marriage is encouraged and acknowledged, with intrafamilial partnerships accounting for 20–50% of all marriages, which is considered a very high percentage compared to other societies [18,73,74,75,76,77].

This review has several limitations. There is diversity [78] among ASCs in many aspects including culture, the economic profile, political stability, research funding and infrastructure, and health care systems. This makes it very difficult to generalise the results of this review across ASCs. Furthermore, differences in methodology, including hospital versus community sittings, in the included studies also contributed to high levels of heterogeneity in the results limiting generalisability. Despite these limitations, these meta-analyses provide a snapshot of CP among children and adolescents in ASCs. Further research is required to adequately represent and include all countries in the region, to provide a clearer picture of the epidemiology of CP in ASCs.

This is important data that is useful for planning services for children with CP in ASCs. A large number of evidence-based interventions exist for children with CP, which may improve motor outcomes, physical activity, and participation; manage tone and dysphagia; prevent contracture and secondary impairments such as hip displacements; and provide targeted early intervention and intervention for parents [79]. Such interventions should be goal-based and consider the activity and participation levels, as well as the body structures and function levels, of the International Classification of Functioning, Disability and Health (ICF).

## 5. Conclusions

This systematic review and meta-analysis provides novel and timely knowledge informing our understanding of CP in ASCs. However, the epidemiology of CP among children and adolescents in ASCs is in dire need of further research. In recent years, the number of CP registers globally has grown significantly. Registers provide a platform for epidemiological research, a sampling frame for clinical research, and can inform health service planning and provision. There is an excellent example of the feasibility of this type of register in ASCs, which is the CP register in Jordan [80]. The establishment of similar registers in other ASCs has great potential to expand our understanding of the epidemiology of CP and improve the lives of people living with CP and their families.

## Figures and Tables

**Figure 1 brainsci-12-00859-f001:**
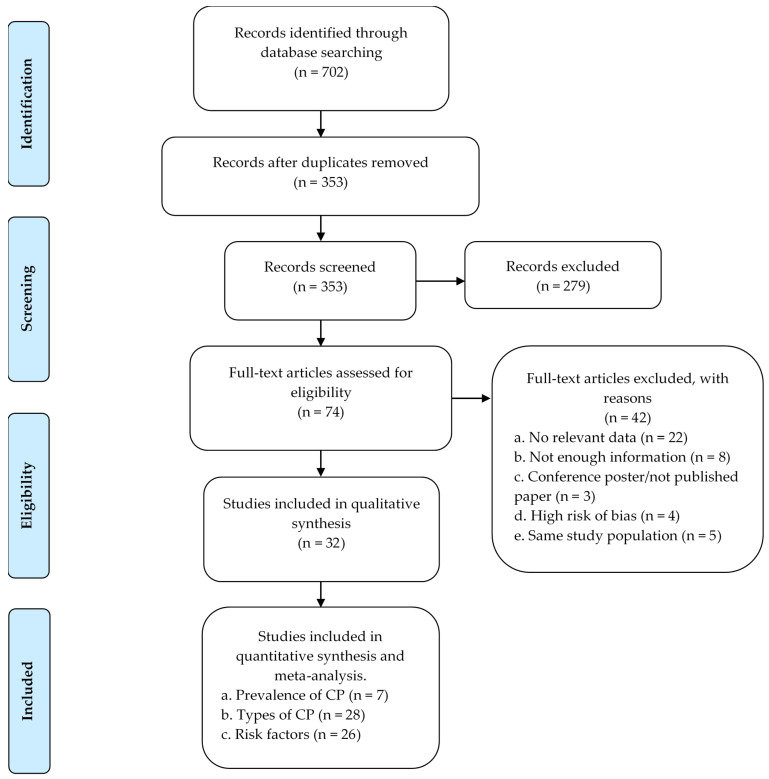
PRISMA flow diagram for a systematic literature review and study selection; CP among children and adolescents in ASCs.

**Figure 2 brainsci-12-00859-f002:**
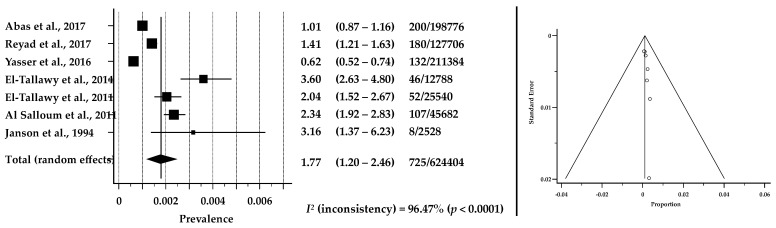
Forest plot (to the left) shows the pooled estimates of prevalence of CP among children and adolescents in Arabic-speaking countries [31,46,47,48,49,50,51,52]. Assessment of publication bias by Funnel plot (to the right) shows individual study estimates against corresponding standard errors (Kendall’s Tau = 0.4286, *p* = 0.1765). *I*^2^ suggests considerable heterogeneity between studies.

**Figure 3 brainsci-12-00859-f003:**
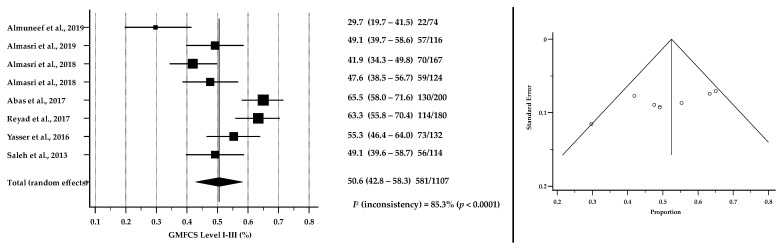
Forest plot (to the left) shows the pooled estimates % of GMFCS level I-III among children and adolescents with CP in ASCs [29,39,40,41,42,47,48,49]. Assessment of publication bias by Funnel plot (to the right) shows individual study estimates against corresponding standard errors (Kendall’s Tau = −0.5000, *p* = 0.0833). *I*^2^ suggests considerable heterogeneity between studies.

**Figure 4 brainsci-12-00859-f004:**
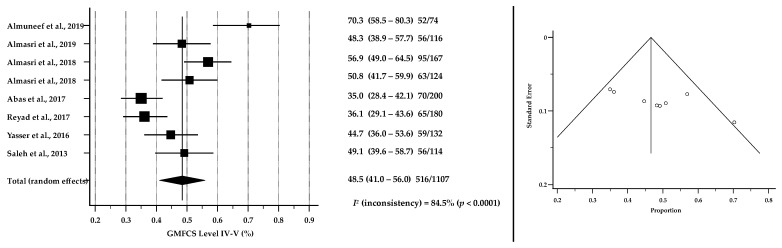
Forest plot (to the left) shows the pooled estimates % of GMFCS level IV-V among children and adolescents with CP in ASCs [29,39,40,41,42,47,48,49]. Assessment of publication bias by Funnel plot (to the right) shows individual study estimates against corresponding standard errors (Kendall’s Tau = −0.5714, *p* = 0.0478). *I*^2^ suggests considerable heterogeneity between studies.

**Figure 5 brainsci-12-00859-f005:**
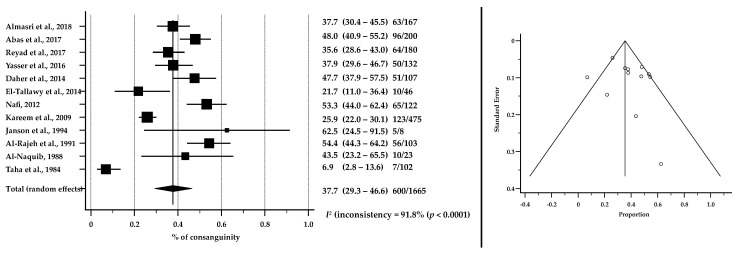
Forest plot (to the left) shows the pooled estimates % of consanguinity among children and adolescents with CP in ASCs [35,36,38,40,43,46,47,48,49,50,55,60]. Assessment of publication bias by Funnel plot (to the right) shows individual study estimates against corresponding standard errors (Kendall’s Tau = 0.09091, *p* = 0.6808). *I*^2^ suggests considerable heterogeneity between studies.

**Table 1 brainsci-12-00859-t001:** Eligible publications per country included in this review. The table also shows the classification of ASCs by income and child population of each country.

#	Country/State ^1^	Classification by Income	Children and Adolescents Aged 0–19 Years ^2^ N (% of Total Population)	Studies Included (N)
1	Algeria	LMIC	16,409,237 (37)	–
2	Bahrain	HIC	399,990 (24)	–
3	Comoros	LMIC	428,906 (49)	–
4	Djibouti	LMIC	376,430 (38)	–
5	Egypt	LMIC	43,413,971 (42)	6
6	Emirates	HIC	1,854,704 (19)	–
7	Iraq	UMIC	19,320,987 (48)	4
8	Jordan	UMIC	4,392,416 (43)	8
9	Kuwait	HIC	1,141,552 (27)	–
10	Lebanon	UMIC	2,287,154 (34)	–
11	Libya	UMIC	2,471,165 (36)	1
12	Mauritania	LMIC	2,315,383 (50)	–
13	Morocco	LMIC	12,849,811 (35)	–
14	Oman	HIC	1,362,877 (27)	–
15	Palestine ^3^	LMIC	2,474,021 (48)	1
16	Qatar	HIC	498,936 (17)	–
17	Saudi Arabia	HIC	10,816,497 (31)	10
18	Somalia	LIC	9,152,954 (58)	–
19	Sudan	LIC	22,252,463 (51)	2
20	Syria	LIC	6,961,028 (40)	–
21	Tunisia	LMIC	3,657,697 (31)	–
22	Yemen	LIC	14,783,682 (50)	–
Total (N) studies				32

ASCs: Arabic-speaking countries; N: number; LIC: Low-Income Country; LMIC: Lower-Middle Income Country; UMIC: Upper-Middle-Income Country; HIC: High-Income Country. ^1^ Ordered alphabetically. ^2^ The 2020 estimates according to Department of Economic and Social Affairs, United Nations [21]. ^3^ West Bank and Gaza [22].

**Table 2 brainsci-12-00859-t002:** Characteristics of included publications (alphabetically ordered by country name).

	First Author, Year	Country (Economic Profile) ^1^	Design	Setting	Population	Study Period (Year) ^2^	Children with CP (N)	Participants’ Age in Years (Mean) Range ^3^	Male %
1	Reyad et al., 2017 [47]	Egypt (LMIC)	Cross-sectional	Hospitals and rehabilitation centres	Children with CP	7 months	180	(NR) 0.5–15	58.0
2	Abas et al., 2017 [48]	Egypt (LMIC)	Prospective cross-sectional; individual assessment	Rehabilitation centres	Children with CP	7 months (2015)	200	(NR) 0.25–18	55.0
3	Yasser et al., 2016 [49]	Egypt (LMIC)	Cross-sectional	Hospitals and rehabilitation centres	Children with CP	No information	132	(NR) 0.5–15	59.1
4	El-Tallawy et al., 2014 [50]	Egypt (LMIC)	Cross-sectional; door-to-door; with comparison group	Community	All residents	No information	46	(10.3) NR	66.7
5	El-Tallawy et al., 2014 [51]	Egypt (LMIC)	Cross-sectional, descriptive, population-based, and case-control	Community	Children with CP	No information	98	(7.9) NR	53.1
6	El-Tallawy et al., 2011 [52]	Egypt (LMIC)	Cross-sectional; door-to-door	Community	All residents	17 years (1990–2007)	52	(7.17) NR	44.2
7	Khadir et al., 2020 [53]	Iraq (UMIC)	Case-series	Hospital	Children with CP	8 months (2016–2017)	48	(1.8) <1–5	70.8
8	Salman et al., 2019 [54]	Iraq (UMIC)	Case-control	Hospital	Children with CP	5 months (2012–2013)	100	(NR) ^4^ 0.25–13	58.0
9	Kareem et al., 2009 [55]	Iraq (UMIC)	Case-series	Hospital	Children with CP	40 months (2005–2008)	475	(6.67) 0.83–12.5	63.8
10	Hassan, 2009 [56]	Iraq (UMIC)	Case-series	Rehabilitation centre	Children with CP	1 year (2002–2003)	100	(NR) 0–1.5	57.0
11	Almasri et al., 2019 [39]	Jordan (UMIC)	Cross-sectional	Hospital	Children with CP	No information	116	(4.6) NR	53.4
12	Almasri et al., 2018 [40]	Jordan (UMIC)	Population-based case-series	Hospitals, rehabilitation centres, and schools for children with CP	Children with CP	2 years (2013–2015)	167	(3.6) NR	58.3
13	Almasri et al., 2018 [41]	Jordan (UMIC)	Population-based case-series	Hospitals, rehabilitation centres, and schools	Children with CP	No information	124	2–16 (4.5)	55.6
14	Saleh et al., 2013 [42]	Jordan (UMIC)	Case-series	Hospitals	Children with CP and their families	No information	114	(NR) 0.08–17	53.5
15	Nafi, 2012 [43]	Jordan (UMIC)	Case-series	Early Diagnostic Centre	Children with CP	32 months (2007–2010)	122	(6.3) 0.58–17	54.1
16	Al-Ajlouni et al., 2008 [44]	Jordan (UMIC)	Case-series	Hospital	Children with CP	2 years (2006–2007)	158	No information	53.0
17	Al Ajlouni et al., 2006 [45]	Jordan (UMIC)	Cross-sectional; individual assessment	Hospital	Children with CP	15 years (1990–2005)	200	(3.19) 0.08–15	69.0
18	Janson et al., 1994 [46]	Jordan (UMIC)	Cross-sectional	Community	All children	No information	8	(NR) 0–7	50.6
19	Khan, 1992 [59]	Libya (UMIC)	Case-control	Hospital	Children with CP	4 years (1983–1987)	60	(NR) 1–15	65.0
20	Daher et al., 2014 [60]	Palestine (LMIC)	Case-control	Hospitals and rehabilitation centres	Children with CP	8 months (2011)	107	(3.87) 1–15	54.2
21	Almuneef et al., 2019 [29]	Saudi Arabia (HIC)	Cross-sectional; individual assessment	Rehabilitation centre	Children with CP	8 months (2015)	74	(5.6) 1–12	59.5
22	Abolfotouh et al., 2018 [30]	Saudi Arabia (HIC)	Retrospective/prospective cohort	Hospital	New-borns with extreme low birth weight (ELBW)	3 years (2005–2007)	23	Infants	56.4
23	Al Salloum et al., 2011 [31]	Saudi Arabia (HIC)	Cross-sectional	Community	All residents	2 years (2004–2005)	107	(NR) 0–19	61.7
24	Al-Asmari et al., 2006 [32]	Saudi Arabia (HIC)	Case-series	Hospital	All children	20 years (1984–2003)	412	(NR) 1–10	62.6
25	Al-Sulaiman et al., 2003 [33]	Saudi Arabia (HIC)	Case-series	Hospital	Children with CP	1 year (2000)	187	(1.7) 1–3	58.3
26	Izuora et al., 1992 [34]	Saudi Arabia (HIC)	Case-series	Hospital	Children with neurological problems	18 months (1988–1990)	34	(4.4) 0–12 ^5^	57.2
27	Al-Rajeh et al., 1991 [35]	Saudi Arabia (HIC)	Case-control	Hospital	Children with CP + control group	5 years (1984–1988)	103	(3.8) 1–12	55.3
28	Al-Naquib, 1988 [36]	Saudi Arabia (HIC)	Case-series; individual assessment	Hospital	Children with neurological problems	1 year (1984–1985)	23	(NR) 0.08–14	35.0
29	Alfrayh et al., 1987 [37]	Saudi Arabia (HIC)	Retrospective chart review	Hospital	Children with neurological problems	18 months (1982–1983)	52	(NR) 0.25–15	36.5
30	Taha et al., 1984 [38]	Saudi Arabia (HIC)	Case-series	Hospital	Children with CP	3 years (1980–1983)	102	(NR) 2–9	59.8
31	Salih, 2020 [57]	Sudan (LIC)	Retrospective hospital-based	Hospital	Children with CP	3 years	108	No information	54.6
32	Abdullahi et al., 2013 [58]	Sudan (LIC)	Case-control	Hospital	Children with CP	6 months (2012)	111	(4.1) 1–11	53.2

CP: cerebral palsy; M: male; F: female; ELBW: extremely-low-birth weight, N: number. ^1^ LIC: Low-Income Country; LMIC: Lower-Middle Income Country; UMIC: Upper-Middle-Income Country; HIC: High-Income Country. ^2^ We mentioned periods and years when available. ^3^ In order to unify all units, we converted months to years. ^4^ NR: not reported. ^5^ Range from 2 days to 12 years.

**Table 3 brainsci-12-00859-t003:** Meta-analysis of CP types among children and adolescents in ASCs.

Types of CP	Number of Included Children ^1^	Total % (Random Effects)	95% CI	Heterogeneity *I*^2^ (Inconsistency) %	Publication Bias (Kendall’s Tau)
Spastic (all)	2099	59.8	46.2–72.7	98.6	−0.2523; *p* = 0.0595
Quadriplegic	1024	25.1	18.2–32.8	96.2	−0.01328; *p* = 0.9210
Diplegic	778	16.9	11.4–23.3	95.7	−0.1301; *p* = 0.3311
Hemiplegic	436	10.4	7.3–13.8	90.1	−0.03453; *p* = 0.7965
Dyskinetic/Athetoid	218	4.4	2.9–6.2	82.1	−0.2899; *p* = 0.0304
Ataxic	121	2.7	1.5–4.2	83.2	−0.04515; *p* = 0.7360
Hypotonic/Atonic	228	4.1	1.7–7.3	94.4	0.1089; *p* = 0.4161
Mixed	171	4.6	2.3–7.7	93.0	0.3958; *p* = 0.0031
Unclassified	292	3.7	1.5–6.9	94.8	0.1463; *p* = 0.2747

CP: cerebral palsy; ASCs: Arabic-speaking countries; CI: confidence interval. **^1^** The sum of the numbers here is not a total sum, as some studies reported spastic only and some reported spastic subtypes. The statistics have been conducted independently for each group.

**Table 4 brainsci-12-00859-t004:** Publications that reported CP cases according to Gross Motor Function Classification System.

	Country (Economic Profile)	Setting	GMFCS N (%)							
			Level I	Level II	Level III	Level I–III	Level IV	Level V	Level IV–V	Unclassified
Almuneef et al., 2019 [29]	Saudi Arabis (HIC)	Rehabilitation centre	-	-	-	22 (29.7)	-	-	52 (70.3)	-
Almasri et al., 2019 [39]	Jordan (UMIC)	Cross-sectional	19 (16.4)	14 (12.1)	24 (20.7)	57 (49.1)	31 (26.7)	25 (21.6)	56 (48.3)	3 (2.6)
Almasri et al., 2018 [40]	Jordan (UMIC)	Population-based case-series	18 (10.7)	33 (19.6)	19 (11.3)	70 (41.7)	51 (30.4)	44 (26.2)	95 (56.6)	-
Almasri et al., 2018 [41]	Jordan (UMIC)	Population-based case-series	16 (13.1)	28 (23.0)	15 (12.3)	59 (48.4)	41 (33.6)	22 (18.0)	63 (51.6)	-
Abas et al., 2017 [48]	Egypt (LMIC)	Prospective cross-sectional; individual assessment	19 (9.5)	45 (22.5)	66 (33.0)	130 (65.0)	39 (19.5)	31 (15.5)	70 (35.0)	-
Reyad et al., 2017 [47]	Egypt (LMIC)	Cross-sectional	11 (6.1)	40 (22.2)	64 (35.6)	115 (63.9)	55 (30.5)	10 (5.5)	65 (36.1)	-
Yasser et al., 2016 [49]	Egypt (LMIC)	Cross-sectional	14 (10.6)	23 (17.4)	36 (27.3)	73 (55.3)	20 (15.2)	39 (29.6)	59 (44.7)	-
Saleh et al., 2013 [42]	Jordan (UMIC)	Case-series	18 (15.8)	14 (12.3)	24 (21.1)	56 (49.1)	31 (27.2)	25 (21.9)	56 (49.1)	2 (1.8)

CP: cerebral palsy; GMFCS: gross motor function classification system; N: number. Shaded columns are for clarification of combined groups.

**Table 5 brainsci-12-00859-t005:** Meta-analysis of risk factors of CP among children and adolescents in ASCs.

Risk Factors	Total % (Random Effects)	95% CI	Heterogeneity *I*^2^ (Inconsistency) %	Publication Bias (Kendall’s Tau)
Consanguinity ^1^	37.7	29.3–46.6	91.8	0.09091; *p* = 0.6808
Unknown	30.7	20.3–42.2	98.0	0.1113; *p* = 0.4253
Birth asphyxia	16.0	8.8–24.9	97.6	0.07419; *p* = 0.5951
Preterm	12.2	6.9–18.7	96.5	−0.1731; *p* = 0.2150
Birth weight <2500 gm	9.7	4.4–16.8	97.4	0.006182; *p* = 0.9647
Neonatal Jaundice	5.9	1.9–11.9	97.5	0.3277; *p* = 0.0189
Acquired infections	3.3	1.4–6.1	93.3	0.2040; *p* = 0.1439
NICU admission	2.5	0.4–6.5	97.0	0.6749; *p* < 0.0001
Family history of CP ^2^	2.2	1.0–4.1	89.0	0.5209; *p* = 0.0002
Multiple birth ^3^	1.9	0.6–3.9	91.5	0.3158; *p* = 0.0237
Medical condition of mother ^4^	1.8	0.5–3.9	93.2	0.5139; *p* = 0.0002
Congenital infections ^5^	1.3	0.4–2.8	89.3	0.6059; *p* < 0.0001
Injury/trauma	0.9	0.3–2.0	84.7	0.6502; *p* < 0.0001
Assisted reproduction	0.2	0.1–0.4	0.0	0.9068; *p* < 0.0001

CP: cerebral palsy; ASCs: Arabic-speaking countries; CI: confidence interval; NICU: neonatal intensive care unit. ^1^ Consanguinity was not reported in all studies included in this meta-analysis. We performed an independent meta-analysis for eligible studies that reported this unique risk factor (Section 3.6. below); ^2^ family history of having another child with CP in the same family or relatives; ^3^ e.g., twins; ^4^ e.g., hypertension, diabetes mellitus, and heart problems; ^5^ TORCH infections.

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
