# Peer review of "Epidemiology of Cerebral Palsy among Children and Adolescents in Arabic-Speaking Countries: A Systematic Review and Meta-Analysis"

_brainsci, 2022, doi:10.3390/brainsci12070859_

Round 1

Reviewer 1 Report

Dear authors

·        Thanks for presenting a much needed topic on cerebral palsy epidemiology in the Arab world. The study was well-designed and presented. Although, the results have highlighted a significant knowledge gap, it remains an important knowledge base to build upon in future research.

·        A remarkable feature of this research is that is it shed light on the numerous limitations that may influence the interpretation of results. For example, the relative paucity of participants in this systematic review -pooled total of 3,943 cerebral palsy (CP) children/adolescents- is likely a factor of low research output and not paucity of scientific material/patients. This is closely linked to the substandard health/hospital registration systems especially (electronic records) across parts of the Arab world.

·       The manuscript also touched on the classification of Arab countries according to income (Table 1). In that respect, Egypt the most populous Arab country is a LMIC while most of the Arab countries classified as HIC have the lowest populations and hence lowest numbers of CP children/adolescents available for research enrollment. However, the most advanced hospital registration systems and research infrastructure is mostly available in these under-populated HIC Arab countries. This creates a disadvantaged research milieu in the Arab world in which research output may not be truly reflective of the “Overall” CP epidemiology.

·        Remarkably, study participants from Jordan (N=1009), one of the least populated Arab countries, exceed study participants from Egypt (N=708), the most populated Arab country (Table 2). The percentage of children/adolescents of total population in Egypt is more than 10 times greater than those in Jordan (Table 1).

·        The above point can be attributed to the presence of CP register in Jordan which the manuscript praised in the conclusion. It also supports the notion that national CP registers are critical to uncovering the epidemiological features of CP, boosting research output and subsequent health planning. 

·        Table 1 “Children aged 0 – 19 years N (% of total population)”. It might be more appropriate to consider revising the column heading to Children and adolescents. Children’s age limit typically ends at ~ 11 years.  

Thanks again to the authors for addressing this critical research point.

Author Response

Dear reviewer

Thank you very much for your valuable comments.

The paucity of participants in this systematic review was a challenge because of scarcity of available data. This makes it difficult to generalise the results of this research to all Arab countries, and this was discussed in the paper. However, we agree with you that this research is very important and it will be a substantial basis for much future research in this field. As you mentioned in your valuable comments, the presence of a registry in Jordan for cerebral palsy played a prominent role in the availability of data, which gives strong evidence of the importance of having this type of records to know the true picture of cerebral palsy in the Arab region. This is what we have already concluded at the end of this study and what we strongly advocate.

Regarding Table 1 “Children aged 0 – 19 years N (% of total population)”. It is included children and adolescents. The heading will be updated accordingly. Thank you.

Kind regards,

Sami Mushta (on behalf of authors)

Reviewer 2 Report

The manuscript investigated the prevalence of cerebral palsy among children and adolescents in Arabic-speaking countries through a a systematic review and meta-analysis. The manuscript is very important especially for Arabic countries that lack of the required information for cerebral palsy. I would like to thank the authors for this well-written, and organized work. Please find my few minor following comments: 

1- You excluded 4 articles at high risk of bias. Please define the high risk of bias and justify the exclusion. If possible, compare the results of those articles with your study results. 

2- Please provide the implications based on your findings that may help in health strategies and cerebral palsy protocols. 

3- Table 2: I suggest to sort the studies according to the country rather than the author name 

Author Response

Dear reviewer

Your comments are a great addition to our work, thank you very much.

Despite the difficulty of generalising it to all Arab countries, we believe it is an important addition to the research literature and an important basis for beginning to understand cerebral palsy in this region of the world.

Regarding comment 1, the definition for unacceptably high ROB is included in the methods “2.3. Quality assessment (risk of bias assessment)”. The excluded articles were reported in Appendix B, with their specific risk of bias and reasoning for exclusion. We have amended Appendix B to state this information more clearly.

Regarding comment 2, this paper provides some early data with implications regarding the support of children living with CP in ASCs. We have adapted the discussion to comment on these implications. Thank you.

Regarding the comment 3, your suggestion is appreciated. This will be addressed, thank you.

Kind regards,

Sami Mushta (on behalf of authors)

Reviewer 3 Report

This manuscript fills important knowledge gaps. Analyses largely appear to be appropriate and well-described. I have relatively minor concerns and recommendations.

-Prevalence estimation: In estimating prevalence, Authors include hospital-based studies as well as community-based studies. It is not obvious to this Reviewer that prevalence estimates based on these two types of studies can be easily and meaningfully compared-- e.g., should prevalence estimates from hospital-based studies be considered lower bounds? Please comment on the decision to pool results from these two types of studies.

-Limited data: Authors appropriately indicate (e.g. in Table 1) that extant studies unevenly represent specific nations and populations and appropriately discuss how under-representations may be influencing reported results. While Authors performed funnel plots, potential biases identified in some meta-analyses (e.g. GMFCS IV-V rate estimates) have not been similarly discussed.  Also, raw Tau values are reported in Table 5, but significance values are not reported-- making assessment of bias in these analyses difficult.

Author Response

Dear reviewer

Thanks very much for your important comments. These comments are a great addition to our work.

For the prevalence estimation, we completely agree that there are differences between hospital-based and community-based studies, and complexities with pooling these. We debated this and carefully considered the details of the eligible studies before electing to pool the data. We mentioned the inclusion of both types of studies in the result (section 3.3) stating that We acknowledge that care must be taken when interpreting our pool results, as we included data from both hospital and community-based studies, due to the availability of data reported about prevalence. However, this was important as to elaborate the possible estimation from available data.” And in discussion (in paragraph 2) stating that There is variation in the methodology (i.e., community-based versus hospital-based) and the economic profile of these countries (5 from LMIC, 1 from HIC, and 1 from UMIC). This is reflected in the range of prevalence reported (0.62 to 3.60 per 1000 LB). This gives us an indication of the range of CP birth prevalence across the region but more research is required.”, however have now expanded on this further in the limitations This makes it very difficult to generalise the results of this review across ASCs. Furthermore, differences in methodology, including hospital versus community sittings, in the included studies also contributed to high levels of heterogeneity in results limiting generalisability”. The manuscript has amended and revised. Thank you.

For your comment about limited data, we have amended the manuscript accordingly. Thank you.

Kind regards,

Sami Mushta (on behalf of authors)